# A Phenomenological Investigation into Cyberbullying as Experienced by People Identifying as Transgender or Gender Diverse

**DOI:** 10.3390/ijerph19116560

**Published:** 2022-05-27

**Authors:** Sophie Evelyn, Elizabeth M. Clancy, Bianca Klettke, Ruth Tatnell

**Affiliations:** School of Psychology, Deakin University, Geelong 3216, Australia; smackenzie@deakin.edu.au (S.E.); bianca.klettke@deakin.edu.au (B.K.); ruth.tatnell@deakin.edu.au (R.T.)

**Keywords:** transgender, gender diverse, cyberbullying, qualitative study, adult

## Abstract

Cyberbullying can present a serious risk for adolescents and young adults, with severe effects on victims including adverse mental health outcomes and increased risk of suicide. Transgender and gender diverse (TGD) individuals are significantly more likely to experience cyberbullying. However, little is presently known about the lived cyberbullying experiences of TGD adults despite the prevalence of cyberbullying experienced by the TGD community. TGD participants (*n* = 66, *M* = 24 years) were recruited through snowball sampling and completed an online questionnaire as part of a larger study, which included qualitative questions regarding cyberbullying. Participants reported that cyberbullying was experienced predominantly on social media sites and was largely anonymous. The content was often physically and sexually threatening and heavily transphobic. Additionally, some cyberbullying experienced by TGD individuals was perpetrated by other TGD individuals and focused on the identity policing and gatekeeping of TGD spaces. Participants reported cyberbullying evoked negative emotions, and they often responded by either arguing with or blocking the perpetrators, thereby demonstrating resilience. Some participants sought social support in response to cyberbullying, which acted as a protective factor. Findings reveal specific harms associated with cyberbullying as experienced by TGD individuals and highlight the need for further research and targeted support.

## 1. Introduction

Transgender and gender diverse (TGD) individuals are among the most marginalized groups in modern society [1], and they experience a range of discrimination and abuse, both in person and online. One such online discrimination is cyberbullying, and many TGD individuals have experienced cyberbullying due to a lack of societal acceptance of their gender minority status [2,3]. However, within cyberbullying research, investigations of TGD individuals’ lived experiences are limited. TGD individuals are often included under a broad category of LGBTQIA+, and the vast majority of current research into cyberbullying is limited to predominantly adolescent populations [2,3], despite cyberbullying being prevalent among young adults [4,5].

Both transgender and gender diverse individuals experience pressures attributed to their existence outside cultural gender norms that are heavily based on strict adherence to binary gender roles [6]. Broadly, transgender people identify as a gender different from their sex assigned at birth, whether they medically transition or not, and there is a broad variety in transgender experiences [7,8]. Gender diverse is a collective term used to encompass a variety of gender identities that exist outside binary cultural gender expectations and includes those that identify as non-binary, genderqueer, and agender [6,9,10,11]. In contrast, the term cisgender refers to individuals whose sex assigned at birth is congruent with their gender identity [12].

It is well-established that sexual and gender minorities experience higher levels of distress than their cisgender and heteronormative counterparts [7,13,14]. Even within the LGBTQIA+ spectrum, TGD individuals are disproportionately impacted by mental illness, and they experience depression, anxiety, and suicidality at higher rates than cisgender lesbian, gay, bisexual (LGB), and queer individuals [6,7,8,15]. A systematic review of mental illness in TGD individuals found a lifetime prevalence of 42.1% for mood disorders, 26.8% for anxiety disorders, and 14.7% for substance use/abuse disorders, as well as high rates of self-injury and suicidal ideation compared to cisgender peers [16].

Much of the distress faced by TGD individuals arises from experiences of stigma and discrimination related to their gender identity and presentation. Gender-based discrimination is pervasive and can include physical violence, sexual violence, cyberbullying, and economic and employment discrimination [17], as well as microaggressions, intentional or unintentional slights or insults, and misgendering [18]. Gender-based discrimination has a range of negative psychosocial outcomes for TGD individuals and has been found to independently predict attempted suicide [19].

Though often included with binary transgender individuals for research and sampling purposes, non-binary gender diverse people may experience even greater levels of gender-based discrimination and negative mental health outcomes than binary transgender individuals, as their gender identity further defies cultural gender expectations [6,9,10,20]. Given the vulnerability of TGD individuals to discrimination and the potential for dire effects on mental health, the investigation and understanding of potentially discriminatory experiences, in particular cyberbullying, is critical for the welfare of these individuals.

### 1.1. Cyberbullying

Due to its online nature, cyberbullying presents specific opportunities for harassment and discrimination. Cyberbullying has been defined variously across the literature, and there is a lack of definitional consensus, especially as technologies and the digital landscape are ever evolving [21]. However, for the purposes of the current paper, cyberbullying is defined broadly as actions or communications conducted through digital means that cause harm or distress to the recipient [21]. This definition deliberately omits intent, as the nature of gender-based discrimination means that harm can result even without intent [21]. For example, misgendering, even if unintentional, may cause distress to TGD individuals [18].

Though most cyberbullying research has been conducted with adolescents, cyberbullying also occurs within adult populations and has been associated with increased depression, stress, loneliness, and dependence on alcohol [4,5]. Most cyberbullying research that includes TGD individuals is focused on LGBT adolescents rather than on the wider LGBTQIA+ community or TGD individuals specifically [3,22,23,24]. LGBT adolescents are disproportionally more likely to be victims of cyberbullying than their heterosexual and cisgender peers [3,22,23], and sexual-minority youth are victims of cyberbullying at more than twice the rate of their heterosexual peers [24]. Cyberbullying has been found to contribute significantly to psychological distress for both LGBT youth and young adults [25]. In a study of 250 LGBT adolescents, 41% reported being harassed online about their gender identity or expression [26]. These findings highlight the presence of gender-based discrimination in response to perceived deviations from social gender norms and the consequent vulnerability of TGD individuals whose identities do not conform to these norms [7,23].

Additionally, LGBT youth are more likely to experience anonymous forms of cyberbullying than their non-LGBT peers, for whom cyberbullying is frequently an extension of in-person bullying and perpetrated by peers [3]. LGBT youth are also less likely to report cyberbullying experiences for fear of the exposure of their LGBT status and possible discrimination from authorities [3]. Such results suggest fundamental differences in cyberbullying experiences for LGBTQIA+ individuals, as well as additional obstacles to help seeking and prevention of cyberbullying for these individuals.

Conversely, online communities can provide TGD individuals with access to social support and connection with their peers that they might not otherwise have, reducing social isolation [27,28]. Online TGD spaces can facilitate a sense of community for TGD individuals who might otherwise be isolated from the wider TGD community, and these spaces are often the first point of contact for TGD individuals beginning to challenge or explore their gender identity [12,27,28]. Many TGD individuals rely on online friendships and communities as their primary form of social support [12,28]. For TGD individuals, social support is a vitally important source of resilience and a protective factor against the effects of victimization [12,25,28]. Being able to live authentically and true to their gender identity is critical for the mental health and wellbeing of TGD individuals of all ages, and having access to online spaces often provides this opportunity even when this may not be an option in their offline lives [12].

However, for TGD individuals seeking to connect with peers online, observing harmful or negative content regarding TGD people may act as a barrier to online social support [27]. TGD people may also fall victim to lateral violence within the TGD community, which often takes the form of identity policing in which individuals are judged and attacked for aspects of their identity, due to a focus on a perceived correct way to be transgender [27]. As such, many TGD people may be hesitant to engage with potentially helpful online spaces. In sum, online spaces can represent both community and potential danger for TGD individuals, and there is a need for more in-depth research into this area, especially regarding the cyberbullying experiences of TGD adults, which this study seeks to address.

### 1.2. Minority Stress Model

The Minority Stress Model (MSM) argues that minority populations such as TGD individuals are exposed to hostile and stressful social environments because of their minority status, leading to worse mental health outcomes than for cisgender and heteronormative populations [29,30]. The MSM identifies three processes by which minority populations are exposed to minority stress: (1) external or distal environmental stressors, (2) expectations of harm and resulting vigilance, and (3) internalization of negative social attitudes into negative self-perception [29,30]. These processes have a cumulative impact on the mental health of minority populations, permeating all aspects of their lives, including participation in digital communities. Specifically, cyberbullying arising from gender-based discrimination is an external stressor, leading to vigilance and self-protective behavior, such as avoiding social media, and ultimately the potential internalization of these stressors as internalized transphobia and negative self-perception [2,30].

Given the significantly higher rates of self-harm and suicidality in TGD people compared to cisgender people, understanding the nature of the online experiences of TGD adults is critical in developing targeted prevention methods to minimize the harm experienced by TGD individuals in their online experiences [15,16]. Quantitative research suggests TGD experiences of cyberbullying differ from those of cisgender LGB and heterosexual individuals, but this research is externally driven and lacks insight into the lived experiences of TGD individuals [3]. Research regarding TGD individuals that prioritizes their own voices and experiences of cyberbullying allows for a deeper and more nuanced examination of the ways in which cyberbullying impacts TGD individuals and the broader implications of these experiences. Thus, this study aims to examine cyberbullying experiences of TGD individuals by applying a phenomenological approach, addressing gaps in the research, and providing an initial exploration of responses to two research questions: ‘What are the experiences of adult TGD individuals who are victims of cyberbullying?’, and secondly, ‘How does gender identity play a role, if any, in these experiences?’.

## 2. Materials and Methods

### 2.1. Participants

Study participants included 66 TGD adults: 8 transwomen (12%), 13 transmen (20%), 26 gender diverse individuals (39%), 5 agender (8%), 3 men (5%), 4 women (6%), and 7 individuals (11%) identifying with multiple and/or other identities, e.g., demiboy, genderfluid. Ages ranged from 18–34 years (*M* = 24.24 years, *SD* = 3.75). The sample included participants from Australia (*n* = 30), the United States of America (*n* = 13), the United Kingdom (*n* = 10), Canada (*n* = 5), New Zealand (*n* = 3), and four other Western countries (*n* = 5).

### 2.2. Measures

#### 2.2.1. Demographics

Participants answered basic demographic questions, including their age, country of residence, ethnicity, and education level. Participants were asked, “How would you describe your gender identity?”; choices included “transwoman/transfeminine”, “transman/transmasculine”, “gender diverse/gender non-binary/genderqueer/bigender”, “agender”, “man”, “woman”, or the option to define their gender in their own words.

#### 2.2.2. Cyberbullying Victimization

Qualitative data for cyberbullying experiences were collected from responses to the question: “If you have experienced cyberbullying or other negative online experiences, please describe your experiences in your own words, including whether this related to your gender identity or not. How did you feel, and what was your response?”, to which participants provided free-text responses. This question was included in a mixed-methods questionnaire as part of a larger exploratory study investigating online experiences for TGD individuals, including cyberbullying and sexting.

### 2.3. Procedure

After obtaining ethics approval from the Deakin University Ethics Committee (Reference 2018-168), participants were recruited through snowball sampling via posts on social media platforms chosen for their popularity and broad userbases (Facebook, Twitter, Instagram, Tumblr) and flyers in strategic locations. Recruitment was boosted using the paid recruitment site Prolific, whereby participants were paid approximately £2 each. A plain language statement explained the questionnaire and provided contact details. Participants indicated their consent, then completed an online Qualtrics questionnaire. Inclusion criteria required participants to not be cisgender and to be between 18 and 34 years of age, as young adults are at greater risk of cyberbullying than older adults [4,5]. In total, 164 participants commenced the study. Removal of fake/troll responses (*n* = 1), incomplete responses (*n* = 93), and responses outside the targeted age range (*n* = 4) resulted in an analytical sample of 66 participants who responded to the qualitative question.

### 2.4. Analysis

A hermeneutic phenomenological approach [31] was applied to determine the underlying meaning of the participants’ cyberbullying experiences and contextualize them within the greater theoretical framework. In addition to this, we also drew on aspects of descriptive phenomenology to map out and describe the structure of the TGD cyberbullying experience [31,32]. Thematic analysis [33] within a constructivist paradigm was used to reflect the exploratory nature of the study and the phenomenological lens. The first coder (first author) engaged in familiarization with data through repeated reading of responses, generating codes, and arranging them into preliminary themes and subthemes. A second coder (second author) with data familiarity reviewed 20% of the data (selected randomly), and themes and subthemes were reviewed, revised, and refined in discussion between the first and second authors until a consensus was reached. To create a narrative, these themes were then collaboratively reviewed by both coders for overarching patterns of experiences and responses to cyberbullying.

### 2.5. Reflexivity Statement

The first author is a mature-aged PhD student and former secondary teacher who uses she/her pronouns. She has no personal experience of cyberbullying but has witnessed cyberbullying as a cisgender lesbian woman. The second author is a psychologist and, at the time of publication, PhD student and a cisgender bisexual woman who has personally experienced cyberbullying and witnessed the cyberbullying of LGBTQIA+ peers in personal and workplace settings, as well as that of adolescents. Her research focuses on pernicious online behaviors including sexting, sext dissemination, and cyberbullying. As members of the wider LGBTQIA+ community, the first and second authors drew from their own experiences of cyberbullying to approach this research with compassion and sensitivity to the experiences of the TGD participants. The first and second authors undertook regular discussions of their responses to the data throughout the coding and development of themes to manage their responses.

## 3. Results

The 66 participants provided qualitative responses detailing their personal experiences with cyberbullying, with an average length of 75 words. Thematic analysis identified four main themes and fourteen subthemes, presented in Table 1. The first theme related to the sources and platforms of cyberbullying, the second theme related to its content, and the last themes related to practical and emotional responses. Illustrative quotes are provided in italics.

### 3.1. Theme 1: Sources and Platforms

Participants reported cyberbullying across a range of digital platforms and sources, including from anonymous strangers or people within their own community.

#### 3.1.1. Platforms

Social media platforms (e.g., Twitter, Facebook) were cited as the most commonplace for cyberbullying to occur and to be witnessed. Responses indicated that cyberbullying occurred on posts and/or in the comment sections on others’ pages. As one participant explained, *“I have been repeatedly brigaded by TERFs [trans-exclusionary radical feminists] who have actively harassed and threatened me for merely sharing my experience on my personal twitter.” [Participant 16, Transwoman, 25].*

Gaming spaces were also identified as frequent platforms for cyberbullying, either through in-game voice or text chat or in gaming community spaces, e.g., streaming (Twitch) and gaming-focused chat (Discord). As stated by one participant, *“Video games are the worst, especially competitive games […] I sat there for 20 min as people yelled (both on voice and text channels) curses and slurs at me for simply being trans”. [Participant 19, Woman (trans), 21].*

Finally, online dating apps were further potential platforms for cyberbullying. A participant explained that, *“On dating apps people make rude comments about my gender identity and asking very invasive questions about my reproductive system”. [Participant 7, Transman, 21]*. Cyberbullying via direct digital communication such as emails, texts, and direct messages was present but uncommon.

#### 3.1.2. Sources

Most participants reported that perpetrators were either anonymous or unknown to them; however, the second most common source was cyberbullying perpetrated by others in the TGD community. As one participant stated, *“Seeing queer people get outed by other queer people is pretty much a daily thing”. [Participant 48, Agender, 26].* Rarely, participants reported known perpetrators whereby in-person bullying transferred into cyberbullying.

Participants noted personal experiences of targeted cyberbullying, but also reflected on their experiences of indirect cyberbullying, which was directed at others with the same or related gender identities, and which often had negative effects on the observer. As one participant explained, *“Seeing videos and posts with a soul [sic] purpose of invalidating my experiences as a non-binary person makes me feel scared to come out to more people as I don’t know how many people close to me share the same opinions”. [Participant 1, Non-binary, 19].*

### 3.2. Theme 2: The Nature of Cyberbullying

Participants reported a range of content in their experiences of cyberbullying, including transphobia, technology-facilitated violence, and identity policing.

#### 3.2.1. Transphobia

Transphobia permeated almost all reported examples of cyberbullying. Commonly, this included transphobic slurs, with one participant stating, *“I’ve been called a tranny, a ‘sad confused girl who can’t make up her mind about being a boy or girl”. [Participant 3, Gender diverse, 30]*. Ignoring or denying the person’s gender identity was another common experience for respondents, either by the refusal to use correct pronouns and terms or by insistence on reinforcing the victim’s sex assigned at birth. Participants stated:


*“Many people call me “it” or refuse to use my correct pronouns (they/them)”. [Participant 59, Gender diverse, 22]*



*“I was harassed by a group of cisgender boys on the grounds of being trans. They commented on all of my posts and pictures and messaged me misgendering me, calling me a good girl, etc.”. [Participant 30, Man (trans), 22]*


Appearance and perceived lack of conformity to cisgender beauty standards was another focus for transphobic comments. Though most of these comments originated from cisgender perpetrators, some negative comments regarding appearance were made by fellow members of the TGD community. One participant explained, *“A lot of the comments made towards me have been around the way I look. […] you get a lot of gross and derogatory comments around your body and who you are”. [Participant 56, Gender diverse, 19].*

Being outed was another experience shared by some participants, whereby their gender identity was revealed against their will online. As one participant explained, *“[I’ve] been outed as trans online when I wanted to be stealth and had my birth name outed online against my will”. [Participant 3, Gender diverse, 30].*

#### 3.2.2. Intersectional Minority Identities

A common theme among many responses was how cyberbullying focused on intersectional identities, wherein participants who belonged to other marginalized groups in addition to their gender identity often saw attacks on these other aspects of their identity integrated with the transphobia directed towards them. Sexism, homophobia, and racism were common features of the cyberbullying that participants were subjected to in combination with transphobia. As one participant stated, *“I have experienced cyberbullying because I am trans, because I am brown, because I’m a woman. I have been called everything from the N word to faggot to a whore”. [Participant 20, Woman (trans), 34].*

#### 3.2.3. Technology-Facilitated Violence

Many participants reported experiences of cyberbullying in the form of technology-facilitated violence. Though there was a broad spectrum of technology-facilitated violence reported by participants, the majority were either threats of harm, sexualized in nature, or both.

Sexualized forms of technology-facilitated violence included receiving unwanted sexually explicit images and messages, the sexualization of gender identity, threats of sexual violence, and sexual propositions.

The receipt of unwanted sexual images was a common experience, as one participant explained: 


*“I have randomly received nudes, from guys and surprisingly, from girls without my consent. It disgusts me greatly. Some ask, but will still send after I say no”. [Participant 19, Woman (trans), 21]*


Notably, there appeared to be differences in the nature of sexual violence depending on the gender identity of the victim, especially between those assigned female at birth (AFAB) and transwomen. The sexual violence directed at transwomen was primarily focused on the sexualization of their gender identity, whereas the sexual violence directed at those AFAB was focused on attempts to reinforce their sex assigned at birth over their gender identity. As one participant explained, *“I have men telling me that they would fuck me and make me beg to be a chick again […] I get unwanted pictures of penis[es] and messages asking me to beg for it even though I’m engaged to a woman”. [Participant 5, Transman, 24].*

Threats of physical harm were also a common form of technology-facilitated violence seen in responses, including threats of rape, physical assault, and murder. In addition to threats to harm the victim, several participants reported being encouraged to kill themselves. Participants stated:


*“I’ve been sent a lot of rape threats, had men explicitly tell me how they want to rape and beat me, had them sexualise my previous experiences of being sexually abused, and in a few instances I have had threats of corrective rape”. [Participant 34, Agender, 26]*



*“I receive messages from guys on discord telling me to end myself, that I was a disgrace, etc.”. [Participant 19, Woman (trans), 21]*


The frequency and severity of technology-facilitated violence directed at participants often led to a pervasive sense of threat when active in public online spaces, as participants explained:


*“I also rarely post online as I don’t want to draw attention to myself”. [Participant 27, Gender diverse, 29]*



*“I rarely experience cyberbullying or negative online experiences because I’m too anxious to post or comment on things”. [Participant 25, Gender diverse, 23]*


#### 3.2.4. Identity Policing and Gatekeeping

Identity policing was unique to cyberbullying perpetrated by those within the TGD community, whereby participants were accused of “performing their gender identity incorrectly”, for example, not being transgender enough, or being the “wrong kind” of transgender. Relatedly, participants reported gatekeeping. As one participant explained, *“I have been targeted by other trans people for how I look etc., the fact that I model, the fact I post photoshopped photos which gives others unrealistic expectations of transition”. [Participant 15, Transwoman, 34].*

### 3.3. Theme 3: Response to Cyberbullying

Participants displayed resilience by defining the boundaries of their engagement. The most common response to cyberbullying among participants was to ignore or block perpetrators. Some took it further by collecting evidence and reporting the cyberbullying. Participants explained:


*“Honestly, the second I feel the situation is about to become hostile, screenshot what happened, then I block the person or people involved on all fronts, and completely remove myself from the situation” [Participant 12, Gender diverse, 30]*



*“Sometimes TERFs and transphobes are rude to me [...] but I just ignore it, I’m not going to change their mind and they’ll die angry”. [Participant 53, Demiboy, 26]*


Avoidance was also common, by either avoiding a particular platform or online social spaces in general. As one participant explained, *“[I] find myself often leaving a social platform for extended periods to avoid such harassment. In general, I’ve become particularly inactive online”. [Participant 23, Agender, 22]*

Some actively chose to engage with perpetrators, either by arguing back or by attempting to educate them on TGD issues. Participants explained:


*“I roast them until they can’t stand on their own two feet. Is it right? Probably not. Am I angry in the moment and want them to have a taste of their own medicine? Yes”. [Participant 43, Transwoman, 23]*



*“I can feel good when I know I am making a good point or saying something that I think will get through to the person.” [Participant 47, Gender diverse, 26]*


A few participants also reported seeking emotional support from friends both within their community and without in response to their experiences:


*“I reached out to cis allies and trans friends and they stood up for me when comments about me were made public and said kind things to me in private” [Participant 3, Gender diverse, 30]*


### 3.4. Theme 4: Emotional Response to Cyberbullying

Participants identified a broad range of negative emotions invoked by experiences of cyberbullying; these were grouped into two main categories, intrapersonal and interpersonal.

#### 3.4.1. Intrapersonal

Participants identified many intrapersonal emotional responses to the cyberbullying they experienced and witnessed. These internally focused emotions could be categorized into four main intrapersonal emotions: distress, fear, shame, and resignation.

Distress was by far the most common emotional reaction reported by participants, ranging from sad to suicidal. As participants explained:


*“I always feel a little hurt, and sad that people refuse to understand me”. [Participant 59, Gender diverse, 22]*



*“Sometimes it really gets to me and makes me feel suicidal”. [Participant 20, Woman (trans), 34]*


The next most common emotional response was fear. As a participant explained, *“It’s scary to know how much some people will hate you for being different.” [Participant 2, Gender diverse, 22]*

Cyberbullying also invoked feelings of shame and self-hatred among some participants:


*“It made me feel horrible and really made me hate myself more than I ever thought I could. It made me feel like something was wrong with me, in regards to my gender identity”. [Participant 44, Transman, 19]*


A feeling of resignation to the cyberbullying they experienced was also reported by many participants, as a participant explained:


*“Nazis send me anonymous messages calling me slurs, nothing out of the ordinary”. [Participant 6, Transwoman, 18]*


#### 3.4.2. Interpersonal

Interpersonal emotional responses reported were primarily forms of anger and contempt. According to participants:


*“Honestly it used to make me feel very little as it happened so often that I could just laugh it off […] but now I end up feeling pretty angry”. [Participant 34, Agender, 26]*



*“I still see the hate and it makes me feel frustrated, angry, and pessimistic about people and society”. [Participant 64, Transman, 30]*


## 4. Discussion

This study aimed to explore cyberbullying experiences of adult TGD individuals and sought to identify how gender identity featured in these experiences. Transphobia was almost universally the focus of cyberbullying experienced by TGD individuals and underpinned frequent experiences of technology-facilitated violence. Attacks on gender identity were also present in instances of within-community cyberbullying in the form of identity policing.

Consistent with previous research in general adult populations [5], cyberbullying evoked solely negatively valanced emotions for TGD victims and witnesses. Experiences of cyberbullying also prompted some to seek social support, which helped with managing their negative experience. In response to cyberbullying, it was notable that, although many responses focused on blocking and avoiding possible perpetrators, some participants also reported a willingness to take action in response to cyberbullying, either by engaging with the perpetrator or reporting them.

Unlike findings from cisgender populations, but consistent with previous LGBT adolescent studies [3], participants frequently reported that cyberbullying perpetrators were anonymous or strangers. The high prevalence of anonymous cyberbullying may partly be related to the predominance of social media platforms as sites for cyberbullying, given the opportunity such sites can offer for anonymity. Social media was also the primary source for TGD individuals to witness the cyberbullying of others within their community.

The prevalent experience of anonymous cyberbullying targeted at TGD individuals impacted the sense of safety for the victims. Perpetrator anonymity, coupled with the extremity of the technology-facilitated violence, appeared to give TGD individuals a sense of pervasive threat in online public spaces. Many of the emotional responses reported, especially those that were intrapersonal, made it clear that anonymous cyberbullying and the threat it poses negatively impacted the mental health of participants, consistent with the MSM framework [29]. The constant threat of cyberbullying is a clear external stressor leading to vigilance and self-protective behaviors. This also seemed to be internalized as negative emotions and self-perception in this sample, with frequent reports of avoidance of social media in response to cyberbullying and the internalization of shame and self-hatred expressed in several responses [2,30]. Notably, witnessing cyberbullying of another with the same gender identity also had profound effects on TGD individuals and seemed to elicit a similar emotional impact as personally experiencing cyberbullying. Conceptually, this is consistent with MSM explanations, whereby witnessing the cyberbullying of others within your community acts as an external stressor [29,30].

Consistent with previous findings that TGD individuals have a higher lifetime prevalence of online sexual harassment than their cisgender peers [2], participants commonly reported experiences of technology-facilitated violence featuring explicit sexual content or threats of sexual harm. The sexual nature of the cyberbullying was inherently transphobic and largely centered on gender identity in either a direct or indirect manner. Though online harassment was common across a multitude of gender identities, and there were some commonalities such as receiving unwanted images of genitalia, there were notable differences in the role gender identity played in the sexual technology-facilitated violence of those assigned female at birth (AFAB) and transwomen. For TGD individuals AFAB, cyberbullying seemed to represent overt intentions to reinforce sex assigned at birth over personal gender identities, such as threats of “corrective” rape. For these individuals, sexual threats were used to deny their gender identity and to try to reduce them to dehumanized female sex objects.

In contrast, the sexual cyberbullying directed at transwomen was less concerned with denying their gender identity and instead focused on the external sexualization of their gender identity. This sexualization sought to dehumanize and objectify transwomen but was not a reinforcement of their sex assigned at birth, in contrast to the experience of AFAB individuals. As with other expressions of transphobia, these instances of sexual technology-facilitated violence were ultimately gender identity-focused and constituted gender-based discrimination [17].

Consistent with Powell et al. [2], TGD cyberbullying victimization was primarily identity-focused, including gender identity and other intersectional minority identities such as sexuality. This cyberbullying therefore represents a form of gender-based discrimination, as the cyberbullying focuses on perceived deviations from social gender norms [7]. Additionally, this study found within-community victimization, consistent with previous research, and primarily focused on the identity policing and gatekeeping of TGD spaces [27]. Here, cyberbullying focused on reinforcing perceived “correct ways” to be TGD. This cyberbullying was often focused on physical appearances and used as a justification to exclude TGD individuals from online spaces, wherein victims were seen as not meeting the perpetrator’s personal definitions of transgender identity.

A notable difference in experiences of cyberbullying from studies amongst cisgender and heteronormative populations was the prevalence of resignation among the responses despite the high level of threat cyberbullying posed. This resignation was often coupled with interpersonal emotions such as contempt for the perpetrators, as well as anger, both at the general prevalence of cyberbullying for TGD individuals and at the perpetrators themselves. Many of the responses highlighted that cyberbullying is a common occurrence for TGD individuals, noting how inescapable it is for those open with their gender identities online. This is consistent with research into adolescent LGBT populations, whereby some transgender youth indicated that seeing hurtful or harmful content regarding transgender people is common and considered normative, which seems to be even more salient in this adult sample [27]. This may be because adult TGD individuals have had longer to come to terms with their gender identity and place in society. Adult TGD individuals likely have the benefit of greater maturity and life experience than the adolescent LGBT samples used in most previous research, which may account for this difference.

TGD individuals were not solely passive in their responses to cyberbullying, with many actively fighting back against the perpetrators or seeking private and public social support from other members of the community and allies. There was notable resilience shown by TGD individuals in defining the boundaries of their engagement. Some limited their use of social media, at least in the short term. Others showed a willingness to engage with perpetrators, either to educate or to fight back. This resilience was seen despite pervasive resignation, indicating that, despite the frequency with which TGD individuals experience cyberbullying, they continue to push back against perpetrators. This behavior was directed at self-protection and the future mitigation of harm towards themselves and other TGD individuals, indicating a sense of community-mindedness in the face of adversity.

This sense of community is seen even more explicitly in the importance TGD individuals placed on social support to cope with their experiences of cyberbullying. The importance of social support as a protective factor for TGD individuals is consistent with previous research that social support is vital for coping with victimization [28]. This social support came from a mix of sources, both within the community and without, the commonality as an acceptance of the TGD individual’s identity as valid and respected. The ability to live authentically and to have their gender identity acknowledged and accepted by friends, both those TGD themselves and cisgender allies, appeared to lessen the impact of victimization, which is consistent with previous research [12,25].

Although these results are novel and important, this study has some key limitations. Firstly, as the target population represents only a small percentage of the general population, and recruitment was undertaken during periods of COVID-related restrictions, there were challenges in recruitment and distributing the questionnaire to the target population, especially in non-online spaces. Recruitment was almost exclusively conducted online, hence there is a possible sampling bias with regards to the status of the sample: in particular, most responses were received from participants who indicated they were open with their gender identities online. As such, our understanding of possible differences in cyberbullying for those who are not open with their gender identities is limited. It is noted that our questionnaire was only available in English, restricting our responses to English-speaking participants only. As such, though our questionnaire was available internationally, our results are limited to the experiences of English-speaking individuals. Additionally, our qualitative data were collected through responses to a single question on an anonymous questionnaire. Therefore, there was no opportunity for follow-up questions, and significant variations were noted in the amount and detail of information provided.

## 5. Conclusions

This study identifies some important characteristics of cyberbullying experiences for TGD individuals, their perpetrators, and the content and focus of the cyberbullying. This study provides an initial exploration of TGD cyberbullying experiences and supports the need for further qualitative research, particularly a more substantive interview-based qualitative study with a greater scope, for more comprehensive data collection.

Given the severe impact that cyberbullying has on TGD individuals, as evidenced by this study, and given that TGD individuals are already at significantly greater risk of negative mental health outcomes, this vein of research will help to identify and target avenues for the prevention of cyberbullying and support for TGD individuals experiencing cyberbullying, and thus mitigate negative outcomes. Such avenues for future interventions may include legislative changes to address the cyberbullying problem, education programs targeted at reducing gender-based cyberbullying, and avenues for providing TGD individuals greater support in online spaces.

## Figures and Tables

**Table 1 ijerph-19-06560-t001:** Themes and subthemes regarding cyberbullying experiences.

Themes		Subthemes
Sources and platforms	1.	Cyberbullying platforms
	2.	Perpetrators
Nature of cyberbullying	3.	Transphobia
	4.	Intersectional minority identities
	5.	Technology-facilitated violence
	6.	Identity policing and gatekeeping
Responses to cyberbullying	7.	Ignore or block
	8.	Collect evidence and report
	9.	Avoidance
	10.	Engage to educate
	11.	Engage to fight back
	12.	Seek social support
Emotional responses	13.	Intrapersonal
	14.	Interpersonal

## Data Availability

Aggregated data supporting these results can be provided by the authors on request. However, individual responses cannot be provided to protect confidentiality.

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
