# Peer review of "A Phenomenological Investigation into Cyberbullying as Experienced by People Identifying as Transgender or Gender Diverse"

_ijerph, 2022, doi:10.3390/ijerph19116560_

Round 1
Reviewer 1 Report
This is a paper on an interesting and important topic. The authors wrote this paper with novelty and good logical coherence. Below are some suggestions for the authors to consider.
First, there are some outdated references in the introduction and literature review part, hence more recent works need to be cited.
Second, for the methodology part, the authors need to mention their rationales for choosing those platforms to recruit participants. Also, the questions were a bit general for qualitative data collection.
Third, for the results part, the authors didn't interpret each theme in a relatively balanced way, with the interpretation of some themes too long and some are a bit short.
Last, for the conclusion part, the authors are suggested to talk about both theoretical and empirical implications.
Reviewer 2 Report
Thank you so much for the chance to read this manuscript. I appreciated the opportunity to review this paper that tackles an important issue – that of cyberbullying as experienced by trans or gender diverse individuals. Though I have provided numerous constructive comments, I am hopeful that you continue to rework the manuscript as it can serve a useful role in our discipline. In particular, I wanted to learn more about the research from which this project emerged, hoped to see more engagement with phenomenology as a methodological tradition, will encourage you to consider how your findings are written in a phenomenological manner, and ask you to continue to flesh out the implications of your research.
- Abstract
- Your abstract does a wonderful job of introducing the need for the study and what you found through your research. The one point that I may reconsider stating is about how little is known about cyberbullying of TGD adults differ from cisgender populations. Although I don’t disagree with the claim, I don’t believe that your study is truly answering this as the project solely focused on the experiences of trans or gender diverse participants and thus a comparison to cisgender peers is not evident. In my opinion, a different design would be necessary to respond to this gap in the literature.
- Introduction
- I appreciate the way that you open up your paper. However, I would encourage you to consider the framing of stating that “online discrimination…has, at least in part, been attributed to their gender minority status.” Such phrasing makes it sound like it’s their identities that are the cause of this discrimination as opposed to the transphobic and genderist systems in society. You do a good job of making sure to avoid this in subsequent parts of your manuscript, but I did notice it here.
- Cyberbullying
- I really appreciated the way that you do not include intent in your operationalizing of cyberbulling.
- When discussing cyberbullying, I observed you using different acronyms (LGBT and LGBTQIA+). I would encourage you to either be consistent or explain the differences.
- In the section on cyberbullying, you do a good job of highlighting the existing scholarship but it was not evident from your writing how your study was unique and performs an intervention in the literature. Making this clearer would strengthen your work. You do some of this when you talk about the MSM but not in the literature review.
- Minority Stress Model
- The question of “Does gender identity play a role in these experiences?” reads postpositivistic and implies that an absolute “yes” or “no” response is possible. I would encourage you to consider whether this is an appropriate framing in your qualitative study.
- Materials and Methods
- As a reader, it read a bit confusing to have the participant information introduced before you described the actual study and recruitment strategies. I would encourage you to consider reordering this.
- I wanted to learn more about the study itself as you state that the qualitative data was collected from the response to one singular question. Was this the only question in the survey (I’m assuming not)? If not, what was the intention behind the broader survey? What other research questions were guiding the work?
- As a qualitative manuscript, I would like to learn more about what you mean by a descriptive phenomenological approach and why this was the most appropriate methodology for this study. Moreover, the process of analysis does not read uniquely phenomenological. How did phenomenology as a methodological tradition guide your analysis? Some connections to the methodological literature would be necessary here.
- I so value the positionality statements that you offer. What I would have loved to have seen is perhaps one more sentence explaining how these experiences actually shaped your research beyond simply naming that you have or have not experienced certain realities.
- Results
- Perhaps one of the major concerns that I have with this paper is that your findings do not read phenomenological. They read like general qualitative findings. Phenomenologists are interested in learning about what constitutes as the essence of an experience. Your themes do not tell me about the essence of the experience of cyberbullying for TGD people, but rather, read very descriptive. I would encourage you to spend time reflecting on whether or not you achieved the aims of phenomenology or if you employed more of a general qualitative design.
- I would encourage you to use the language of intersecting minoritized identities, as it is not clear what you mean by “intersectional identities.” This is particularly important to consider as many worry about the conflation of intersectionality with intersecting identities as the latter represents an analytical framework that attends to how overlapping systems of power and oppression co-constitute one another to disproportionately affect particular populations in society.
- Concluding Sections
- I think your discussion was really well constructed. You do a good job of making connections to the existing literature. However, I do wish you all had spent more time with your conclusions. As a reader, I was left wondering what this research means for various audiences. You state that it will help identify avenues for prevention and support, but it would be really beneficial if you can actually tease this out more and explain what these could look like.
- Overall
- Your manuscript is really well written, but I did notice that you all use the passive voice a lot throughout the paper. I would encourage you to minimize the instances of passive voice as it was particularly noticeable.
Reviewer 3 Report
Thank you for the opportunity to review ‘A phenomenological investigation into cyberbullying as experienced by people identifying as transgender or gender diverse’. The paper uses an online questionnaire to investigate participant experiences of having been bullied. This is a well-designed study and a generally well-presented paper; consequently I have only a few improvements to suggest; I hope the authors will accept these as ‘friendly amendments’ to an already strong paper.
My first suggestion is to amend the word ‘survey’ to ‘questionnaire’ throughout. This is both for consistency and more accurately to reflect what actually took place. This was not a mixed-methods study (abstract, line 14) as far as I can tell, since the authors describe only qualitative data and analysis, and this must be amended, since it encourages the reader to anticipate quantitative survey data. Qualitative narrative responses are appropriate for this topic, and there is no need to claim otherwise.
The authors were clear about their recruitment process, and perhaps this online recruitment was inevitable during various national Covid lockdowns. That said, I note that all the named countries of origin for participants are all English-speaking (I acknowledge that Canada is partially Francophone). I think this English bias needs to be identified as a limitation (line 475ff). The authors can say very little about cyberbullying in non-English-speaking countries, that is, countries that may have alternative notions of gender identity and social conformity. I would suggest that non-Anglophone experiences might be different (particularly if we consider South/Southeast Asian contexts).
There were only two mentions of resilience in the paper, and while this is understandable, it reinforces the focus on the problems TGD participants face, and less on the positive and resilient response of the participants (Minority Stress theories and models do this also). Problematising participants in the scientific literature reinforces the kind of victimisation of gender and sexually diverse persons. The trans people I know (an admittedly biased personal sampling of people who have significant lived experience) are powerful and resilient. A takeaway message from this study must not only be ‘poor victims’, but that cyberbullies will get pushback. I wonder if the authors would write the findings and discussion sections in quite the same way if they took a resilience/strength-based approach. This leads to my final concern about this paper, that the proposed interventions are targeted at ‘TGD individuals’, which is shutting the door after the horse has bolted. Surely prevention of bullying at the source is the preferred options. It would be good to see some work and creative thinking about how participants think bullying can be prevented in the first instance.
The references appear recent and relevant.
Specific comments: Should line 134 be ‘gives priority’ [or prioritises] rather than ‘precedence’? And why does line 135 say ‘would allow’? Does not this paper do exactly what they
Round 2
Reviewer 2 Report
Thank you for the changes to your manuscript. It has certainly been bettered as a result!